# The Types and Proportions of Commensal Microbiota Have a Predictive Value in Coronary Heart Disease

**DOI:** 10.3390/jcm10143120

**Published:** 2021-07-15

**Authors:** Lin Chen, Tomoaki Ishigami, Hiroshi Doi, Kentaro Arakawa, Kouichi Tamura

**Affiliations:** 1Department of Medical Science and Cardio-Renal Medicine, Graduate School of Medicine, Yokohama City University, Kanagawa 236-0027, Japan; linchen@njmu.edu.cn (L.C.); doihiroshi6120@yahoo.co.jp (H.D.); hiroking@gamma.ocn.ne.jp (K.A.); tamukou@yokohama-cu.ac.jp (K.T.); 2Department of Cardiology, Sir Run Run Hospital, Nanjing Medical University, Nanjing 210029, China

**Keywords:** commensal microbiota, coronary heart disease, systematic review

## Abstract

Previous clinical studies have suggested that commensal microbiota play an important role in atherosclerotic cardiovascular disease; however, a synthetic analysis of coronary heart disease (CHD) has yet to be performed. Therefore, we aimed to investigate the specific types of commensal microbiota associated with CHD by performing a systematic review of prospective observational studies that have assessed associations between commensal microbiota and CHD. Of the 544 published articles identified in the initial search, 16 publications with data from 16 cohort studies (2210 patients) were included in the analysis. The combined data showed that Bacteroides and Prevotella were commonly identified among nine articles (*n* = 13) in the fecal samples of patients with CHD, while seven articles commonly identified Firmicutes. Moreover, several types of commensal microbiota were common to atherosclerotic plaque and blood or gut samples in 16 cohort studies. For example, Veillonella, Proteobacteria, and Streptococcus were identified among the plaque and fecal samples, whereas Clostridium was commonly identified among blood and fecal samples of patients with CHD. Collectively, our findings suggest that several types of commensal microbiota are associated with CHD, and their presence may correlate with disease markers of CHD.

## 1. Introduction

In recent years, coronary heart disease (CHD) has remained the leading cause of death worldwide, while statins and other pharmacological agents for coronary secondary prevention have failed to completely protect people against CHD, despite their widespread use [1,2,3]. Given the unmet need for effective therapies, there has been increasing interest in targeting novel pathways that underlie the pathogenesis of CHD and in establishing a precise system to track its development. Additionally, to observe the progression of the disease, there is an increasingly important clinical value in discovering a predictive biomarker for CHD. Although the molecular mechanisms responsible for the development of CHD are not completely understood, recent studies have highlighted the critical role of commensal microbiota in CHD [4,5,6], with alterations in the gut microbiota being linked to CHD progression [4]. However, a synthetic analysis of the predictive value of specific types of commensal microbes in CHD patients has not yet been performed. In particular, a cross-site comparison of the types of microbes in the blood, gut, and atherosclerotic plaques of these patient is essential, given the high level of variability observed in the microbiota between different subjects and studies [7,8,9,10,11]. Therefore, in this analysis, we aimed to combine the results from published clinical trials to compare the types, proportions, and sources of commensal microbes in CHD. Our findings revealed several types of commensal microbes common to the atherosclerotic plaques, blood, or gut samples in patients with CHD, and the expression of some specific types of commensal microbes could be used as predictive or disease biomarkers of CHD in the future.

## 2. Materials and Methods

### 2.1. Search Strategy

We searched several electronic databases (PubMed, Embase, Web of Science, Cochrane Library, and ClinicalTrials.gov) up until 8 March 2020 for prospective, observational clinical studies that have investigated commensal microbes in patients with CHD. We used broad search terms (Additional File 1) describing aspects of ‘Gastrointestinal Microbiome’ and ‘Coronary Disease’. These terms were used in combination with “AND” or “OR”. This literature review was performed independently by two investigators, with a third resolving any disputes as needed. The detailed search strategy of PubMed: used: (“Coronary Disease”) or (Coronary Diseases) or (Disease, Coronary) or (Diseases, Coronary) or (Coronary Heart Disease) or (Coronary Heart Diseases) or (Disease, Coronary Heart) or (Diseases, Coronary Heart) or (Heart Disease, Coronary) or (Heart Diseases, Coronary) and (“Gastrointestinal Microbiome”) or (Gastrointestinal Microbiomes) or (Microbiome, Gastrointestinal) or (Gut Microbiome) or (Gut Microbiomes) or (Microbiome, Gut) or (Gut Microflora) or (Microflora, Gut) or (Gut Microbiota) or (Gut Microbiotas) or (Microbiota, Gut) or (Gastrointestinal Flora) or (Flora, Gastrointestinal) or (Gut Flora) or (Flora, Gut) or (Gastrointestinal Microbiota) or (Gastrointestinal Microbiotas) or (Microbiota, Gastrointestinal) or (Gastrointestinal Microbial Community) or (Gastrointestinal Microbial Communities) or (Microbial Community, Gastrointestinal) or (Gastrointestinal Microflora) or (Microflora, Gastrointestinal) or (Gastric Microbiome) or (Gastric Microbiomes) or (Microbiome, Gastric) or (Intestinal Microbiome) or (Intestinal Microbiomes) or (Microbiome, Intestinal) or (Intestinal Microbiota) or (Intestinal Microbiotas) or (Microbiota, Intestinal) or (Intestinal Microflora) or (Microflora, Intestinal) or (Intestinal Flora) or (Flora, Intestinal) or (Enteric Bacteria) or (Bacteria, Enteric) and (risk*[Title/Abstract] or risk*[MeSH:noexp] or risk *[MeSH:noexp] or cohort studies[MeSH Terms] or group*[Text Word]). We then used a similar approach with Embase, Web of Science, Cochrane Library, and ClinicalTrials.gov. Following the PICOS (Participants, Interventions, Comparisons, Outcomes and Study design) principle, the key search terms included (P) patients with CHD; (I) detection of the gene of microbiota; (C/O) compare the types of commensal microbiota between the CHD group and the control group; (S) case-control studies or cohort study.

### 2.2. Study Selection

Prospective, observational, controlled studies that assessed changes in populations of commensal microbes were included if they conducted baseline measurements in a population with CHD, including those with atherosclerosis or acute coronary syndrome (ACS) or chronic CHD (defined as a history of myocardial infarction (MI), percutaneous coronary intervention (PCI), coronary artery bypass grafting (CABG), or a diagnosis confirmed through coronary angiography). Publications without detailed data were excluded from this study. When multiple publications were based on the same source study, we included the publication that had the larger sample size or more relative data.

### 2.3. Data Extraction

The following data from each study were extracted: the first author’s name, publication year, country of the conducted study, sample size, population, source of commensal microbes (atherosclerotic plaque or blood or fecal samples), region from which the genetic expression was quantified, and the specific types of commensal microbes demonstrating statistically significant changes in expression, whether increased or decreased, in CHD.

### 2.4. Data Synthesis and Analysis

The commensal microbes demonstrating statistically significant changes in expression in fecal, blood, or atherosclerotic plaque samples from the 16 included studies were sorted and classified according to the degree to which the populations were increased or decreased. Subsequently, an overall comparison of the gut, blood, and atherosclerotic plaque microbiota was performed, and any microbiota demonstrating a change among at least two publications for each of the three sample types was recorded. Furthermore, interstudy comparisons of atherosclerotic plaque, blood, and gut microbiota were also performed, and the microbiota that were common to at least two body sites were recorded.

## 3. Results

### 3.1. Study Selection

The search process and study selection (presented in Figure 1) identified 544 records of interest. Among these, 163 were excluded for being repetitive, and 353 additional articles were excluded from the analysis because they were review articles, published protocols, lab studies, animal studies, or articles deemed not to be of relevance based on their titles and abstracts. The full texts of the 28 remaining articles were obtained. Several studies were subsequently excluded because they did not meet the predefined inclusion criteria, including those with no relevant outcome data (three articles) and those reporting on unrelated topics (two articles). Four studies were excluded due to insufficient information pertaining to the inclusion criteria, and a second article by the same authors was found to be a repetitive report based on a partial dataset. In total, 16 publications, reporting on 16 cohort studies, were selected for inclusion in the analysis [11,12,13,14,15,16,17,18,19,20,21,22,23,24,25,26].

### 3.2. Characteristics of Included Studies

Ultimately, 16 cohort studies, published from 2011 to 2020, were included in the analysis. These studies reported on the characterization of commensal microbe profiles in CHD patients and analyzed changes in commensal microbe populations in fecal, blood, or atherosclerotic plaque samples. These studies are summarized in Table 1.

In total, this systematic review included data collected from 2210 participants. Three of the studies were conducted in the United States of America, three in Japan, one in Finland, one in France, one in Italy, and seven in China. Three studies reported on patients with ACS, 12 included patients with chronic CHD, and one included patients with atherosclerosis. Thirteen studies analyzed changes in commensal microbes in fecal samples, two of which also analyzed changes in plaque samples, one study reported on changes in the microbiota of plaque alone, while two reported on changes in blood alone (Table 1).

### 3.3. Overall Comparison of the Gut, Blood, and Atherosclerotic Plaque Microbiota

We surveyed changes in the atherosclerotic plaque, blood, and gut (feces) bacterial communities to look for commonalities among the studies. The microbiota commonly identified by three publications in patients’ plaque samples were classified as Streptococcus, whereas the microbiota commonly identified among nine papers in patient fecal samples were classified as Bacteroides and Prevotela, regardless of the direction of change (increased or decreased) (Table 2 and Table 3). After taking the trend of change into consideration, we found that Streptococcus was increased in five studies, whereas Lachnospiraceae was decreased in four studies in the fecal samples of patients (Table 4). Unfortunately, we did not observe any similarities in microbiota between two papers in patient blood samples, which may have been due to only two contributing to the comparisons of bacterial populations in the blood (Table 5). Collectively, these results support the notion that several common types of commensal microbiota that exist in the atherosclerotic plaque, blood, and gut (feces) are associated with CHD.

### 3.4. Comparisons of Atherosclerotic Plaque, Blood, and Gut Microbiotas between Studies

One of the main purposes of this study was to search for microbial communities that were commonly observed between gut and atherosclerotic plaque samples, or between gut and blood samples of those with CHD. Although all three sites typically express distinct microbial communities, the comparison of the overall bacterial community compositions revealed commonalities in the microbiota between the atherosclerotic plaque and fecal samples, and between the blood and fecal samples. However, no specific microbial communities were present in all three sample types. Table 6 summarizes the microbial communities that were identified in fecal samples and at least one other sample type in at least two studies. Clostridiales populations were observed in both the blood and fecal samples of patients, whereas Veillonella, Proteobacteria, and Streptococcus communities were identified in atherosclerotic plaque and gut samples in at least two studies. Interestingly, only Streptococcus was reported to have increased in all eight studies. Collectively, these results indicate that several specific types of commensal microbiota coexist in the atherosclerotic plaque and gut or in the blood and gut of patients with CHD.

## 4. Discussion

In this study, we compared the bacterial compositions of the microbiota of the blood, gut, and atherosclerotic plaque from 16 relevant studies of patients with CHD. This approach allowed us to generate a relatively comprehensive description of the microbial communities associated with CHD. This comparison of blood, gut, and atherosclerotic plaque samples was necessary to identify members of the normal microbiome that may translocate from one body habitat to another where they may contribute to disease. We specifically identified several common types of commensal microbiota that existed or coexisted in atherosclerotic plaques, blood, and gut that were associated with CHD. However, we did not observe any similarities between the microbiota populations in the blood and plaque, or between the blood, plaque, and gut of the patients assessed by the 16 included studies, which may have been due to the paucity of relevant studies meeting the inclusion criteria. Our findings suggest that specific types of commensal microbiota, such as Streptococcus, Lachnospiraceae, and Clostridiales, may have a stronger predictive value in CHD. Moreover, the atherosclerotic plaque and blood microbiota may, at least in part, be derived from those present in the gut.

In this study, we found that the expression of Streptococcus was increased in the gut but was also present in atherosclerotic plaque samples; this may represent a previously unappreciated core member of the atherosclerotic plaque communities. This organism is known to be implicated in endocarditis [27]; however, the role of Streptococcus in CHD has not yet been reported. Its increased expression in the gut and its presence in atherosclerotic plaques suggests that it may directly affect the pathogenesis of atherosclerosis. Recently, some studies have revealed that the gut microbiome directly affects immune responses that regulate chronic inflammatory diseases, such as atherosclerosis [4,5], and it is becoming clear that microbiota-derived bioactive compounds can signal to distant organs, contributing to the development of cardiovascular disease states [28]. In addition, the molecular mechanism involving the “molecular mimicry” of microbial antigens has also been found to be associated with atherosclerosis [29]. For example, Binder et al. showed that pneumococcal vaccination decreases the formation of atherosclerotic lesions through a molecular mimicry mechanism between Streptococcus pneumoniae and oxidized low-density lipoprotein (LDL) [30], while our previous clinical research revealed that many autoantibodies that differ from those found in chronic autoimmune diseases are associated with atherosclerosis [31]. In parallel with these findings, a recent study reported that autoantibodies produced by B lymphocytes are present in plaques and may cross-react with the outer membrane proteins of bacteria, as well as with a cytoskeletal protein involved in atherogenesis [32]. Moreover, Saita et al. also demonstrated that B cells present in both the coronary and carotid plaques of patients with cardiovascular diseases locally produce antibodies that are capable of reacting in response to antigens of the gut microbiota and that they may cross-react with self-antigens. Furthermore, immunoglobulin G1 (IgG1) is secreted in human coronary atherosclerotic lesions and recognizes the outer membrane proteins of Enterobacteriaceae [33]. These findings demonstrate that in human atherosclerotic plaques, a local cross-reactive immune response may occur, wherein antibodies cross-react with a bacterial antigen and a self-protein. In addition, antibodies and B lymphocytes could play an important role in these disease processes [31,32].

Besides being present in atherosclerotic plaques, Streptococcus expression was also observed to be simultaneously increased in the gut. Recent studies have found that low levels of microbiota can also enter the bloodstream to systemically induce chronic, low-grade inflammation [3,34]. Generally, the intestinal mucosal barrier plays a critical role in preventing the translocation of bacterial components. This barrier is efficient when the microbiome is complex and stable; however, under certain conditions, such as those induced by diets high in fat and cholesterol or in certain diseases, major alterations to the composition of the host microbiota can occur, which have in turn been associated with increased intestinal permeability [35,36,37,38]. When the intestinal mucosal barrier becomes compromised, commensal microbes or commensal microbe-derived molecules can readily enter the bloodstream and exert systemic effects, which include the induction of infection or chronic low-grade inflammation and immunoreactivity, affecting multiple immune cell populations; this phenomenon has been found to be prevalent in atherosclerosis [39]. However, the presence of Streptococcus in the blood was not identified in at least two of the relevant studies included in our analysis. Recently, another mechanism has been identified through which bacteria could reach the atherosclerotic plaque, which involves phagocytosis by macrophages at epithelial linings (e.g., of the gut and lung). Upon phagocytosis, macrophages become activated; once they reach the activated endothelium of the atheroma, they leave the bloodstream to enter the atheroma and transform into cholesterol-laden foam cells [40]. In support of this mechanism, patients with cardiovascular disease exhibit a two-fold increase in the number of C. pneumonia-infected peripheral blood mononuclear cells compared with that of controls [11]. Furthermore, the bacteria have been shown to only be present in atheromas and not in healthy aortic tissues in mice [41], and they have been identified in human atherosclerotic plaques [42]. Thus, infected macrophages may specifically target bacteria in atheromas. It remains a possibility that Streptococcus can reach atherosclerotic plaques via the systemic circulation and directly promote local inflammatory cascades or elicit a specific immune response, such as molecular mimicry, thereby indirectly influencing host metabolism and systemic inflammation; however, the specific mechanisms by which Streptococcus regulates the development of atherosclerosis remain unknown and require further investigation.

Study Strengths and Limitations

Although some studies investigating the relationship between commensal microbiota and CHD have been conducted previously [6], our synthetic analysis is the first to investigate the types of commensal microbiota that are commonly associated with CHD. Moreover, our findings revealed several specific types of commensal microbiota that commonly exist or coexist in the atherosclerotic plaque, blood, or feces of patients with CHD. This could be valuable knowledge for future studies investigating this association, as it reveals that the microbiota of the atherosclerotic plaque may, at least in part, be derived from the gut and that these specific types of commensal microbiota may have predictive value for CHD.

A limitation of this study was that only 16 relevant publications met the criteria for inclusion, with only three contributing to a comparison of bacterial populations in the atherosclerotic plaque and with only two studies contributing to that in the blood. This limited our ability to generate a precise descriptive summary of the ‘real-world’ changes in relative commensal microbiota populations. Furthermore, there were several differences in the methodology of these studies, such as the region of genetic quantification, which could have influenced the results or may even have been significant sources of potential inaccuracy. In addition, the types and proportions of commensal microbiota could also have been influenced by several patient characteristics, such as diet (including regional differences) [43]. The addition of new evidence to the field will significantly reduce the effects of these limitations in future analyses.

Further studies should investigate specific types of commensal microbiota, as well as factors that modulate or inhibit their activity. In addition, more information is required to verify the predictive value and mechanisms of commensal microbiota in CHD patients.

## 5. Conclusions

In summary, this study revealed key types of bacteria associated with CHD, as well as several types that were simultaneously present in the atherosclerotic plaque, blood, or gut. In addition, the atherosclerotic plaques and blood samples of patients contained numerous bacteria from different phyla. Our findings strongly support the hypothesis that the gut can be a source of atherosclerotic plaque- and blood-associated bacteria. Although our findings are based on the data from a limited number of studies, they clearly suggest that several specific types of commensal microbiota have a predictive value for CHD. More prospective studies are needed to further evaluate this relationship and to identify the mechanisms that drive it. 

## Figures and Tables

**Figure 1 jcm-10-03120-f001:**
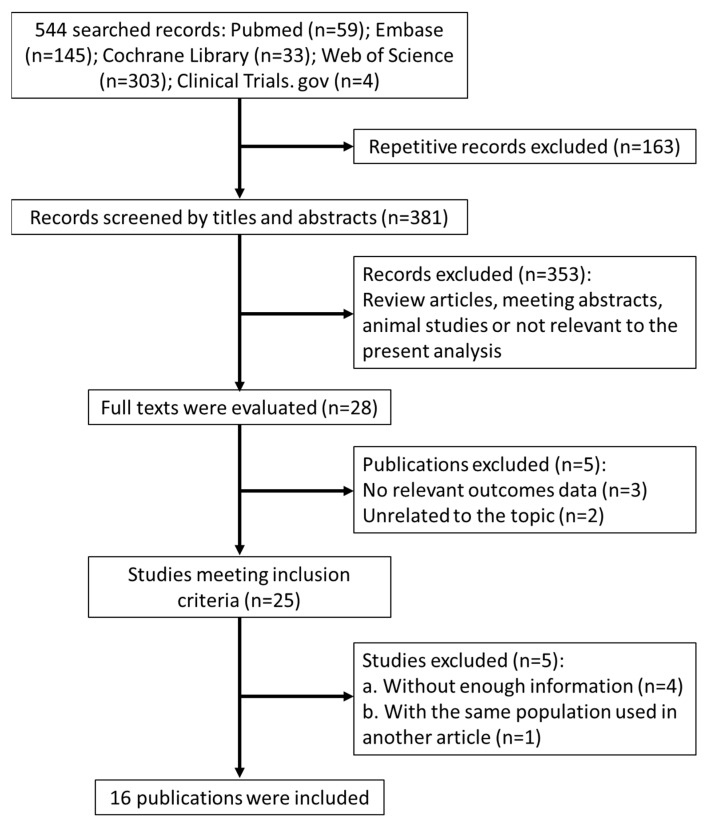
Flow diagram of the study selection.

**Table 1 jcm-10-03120-t001:** Characteristics of included studies.

Study	Country of Origin	Population	Sample Size	Type of Sample			Region of Gene Quantification
				Fecal Samples	Plaque Samples	Blood Samples	
Tuomisto S, 2019 [13]	Finland	CHD	67	√	√		DNA
Liu HH, 2019 [14]	China	CAD	201	√			V3-V4 of 16S rRNA
Emoto T, 2016 [15]	Japan	CAD	119	√			16S rDNA
Cui L, 2017 [16]	China	CHD	64	√			V3-V5 of 16S rRNA
Yoshida N, 2018 [17]	Japan	CAD	60	√			V3-V4 of 16S rRNA
Amar J, 2019 [18]	France	MI	201			√	V3-V4 of 16S rDNA
Emoto T, 2017 [19]	Japan	CAD	69	√			16S rDNA
Toya T, 2020 [20]	USA	Advanced CAD	106	√			V3-V5 of 16S rDNA
Zhu Q, 2018 [21]	China	CAD	168	√			V4 of 16S rRNA
Gao J, 2020 [22]	China	ACS	90	√			V4 of 16S rDNA
Zheng YY, 2020 [23]	China	CAD	309	√			V3-V4 of 16S rRNA
Koren O, 2011 [11]	USA	Atherosclerosis	30		√		V1-V2 of 16S rRNA
Alhmoud T, 2019 [24]	USA	ACS	38	√			V3-V4 of 16S rRNA
Jie Z, 2017 [25]	China	ACVD	405	√			DNA
Li CW, 2016 [26]	China	CAD	206			√	16S rRNA
Pisano E, 2019 [27]	Italy	CAD	77	√	√		16S rRNA

ACS, acute coronary syndrome; ACVD, atherosclerotic cardiovascular disease; CAD, coronary artery disease; CHD, coronary heart disease; MI, myocardial infarction.

**Table 2 jcm-10-03120-t002:** Microbiota commonly identified by at least two publications in the atherosclerotic plaque samples of those with coronary heart disease.

Microbiota in Plaque Samples	No. of Publications (*n* = 3)
Veillonella; Staphylococcus; Burkholderia; Propionibacterium; Corynebacterium; Proteobacteria	2
Streptococcus	3

The column on the right indicates the number of publications for which the microbiota were identified in plaque samples.

**Table 3 jcm-10-03120-t003:** Microbiota commonly identified by at least two publications in the fecal samples of those with coronary heart disease.

Microbiota in Fecal Samples	No. of Publications (*n* = 13)
Enterococcus; Catenisphaera; Coriobacteriaceae; Akkermansla; Veillonella; Erysipelotrichaceae bacterium	2
Proteobacteria; Fusobacteria; Escherichia	3
Lachnospiraceae; Ruminococcaceae; Roseburia; Faecalibacterium	4
Streptococcus	5
Enterobacteriaceae; Lactobacillales	6
Firmicutes	7
Bacteroides; Prevotela	9

The column on the right indicates the number of publications for which the microbiota were identified in fecal samples.

**Table 4 jcm-10-03120-t004:** Changes in microbiota populations in fecal samples of CHD patients.

Microbiota in Fecal Samples	Increase	Decrease	No. of Publications (*n* = 13)
Catenisphaera; Coriobacteriaceae	√		2
Fusobacteria; Escherichia	√		3
Lachnospiraceae		√	4
Streptococcus	√		5

The columns in the middle and on the right indicate the direction of change in the expression of microbiota populations in the feces of patients with coronary heart disease (CHD), and the number of publications reporting the change.

**Table 5 jcm-10-03120-t005:** Microbiota commonly identified in blood samples.

The Change of Microbiota	
Increase	Sphingobacteria, hymenobacter, virgisporangium, micromonosporaceae, bauldia, rhizobiales, Pseudomonadaceae, Rahnella, Serratia, Pseudomonas
Decrease	Caulobacteraceae, Clostridiales, Microbacteriaceae, Neisseriaceae, Brevundimonas, Chryseobacterium, Gordonia, Microbacterium

The columns at the right indicate the microbiota that were found in blood samples from two papers.

**Table 6 jcm-10-03120-t006:** Microbiota populations common to at least two body sites.

Microbiota	Plaque + Fecal	Blood + Fecal
Veillonella	√	
Proteobacteria	√	
Streptococcus	√	
Clostridiales		√

The columns in the middle and on the right indicate the types of microbiota that were found in both the plaque and fecal samples or in the blood and fecal samples of those with coronary heart disease.

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
