# Peer review of "The Types and Proportions of Commensal Microbiota Have a Predictive Value in Coronary Heart Disease"

_jcm, 2021, doi:10.3390/jcm10143120_

Round 1
Reviewer 1 Report
General Comments
- The authors submitted a systematic review of the literature with the purpose of analyzing microbial samples and associating them with coronary heart disease. The topic is certainly important. However, I am concerned by the lack of scientific methodology
Specific Comments
- Including “coronary artery disease” in your search for articles would have probably increased the number of results. In your methodology section you describe that the search was conducted to include “coronary heart disease”. This is confusing since in Table 1 most studies
- The search strategy only includes one date, not a range. Does this mean that all studies in history up until March 2020 were included?
- The methodology of this study does not include any hypothesis, nor statistics. In page 2 lines 70-78, it is described that the 16 selected studies were sorted and classified according to the degree to which the populations were increased or decreased. I am not sure what this means, how it was determined, or what is the significance of this.
- What is a synthetic analysis?
- Page 2, Line 82. States that 163 studies were excluded because they were “repetitive”. No explanation regarding what the authors mean by repetitive. It seems like a very large number of studies excluded.
- Page 5, Line 136. It is not clear what the authors mean by comparisons of microbial communities between different sources. I am assuming the authors were looking to see if the same bacterial species were present in blood/feces, etc. However, this lacks scientific methodology. Therefore, I am not confident on the results and discussion that follows. Too many assumptions without supporting data.
Author Response
Dear Editor:
I wish to submit an revised manuscript for publication in the Journal of Clinical Medicine, titled “The types and proportions of commensal microbiota have predictive value in coronary heart disease” The paper was coauthored by Lin Chen , Tomoaki Ishigami , Hiroshi Doi, Kentaro Arakawa and Kouichi Tamura.
We performed point by point responses to the reviewer’s comments as followings;
We believe that our study makes a significant contribution to the literature. Further, we believe that this paper will be of interest to the readership of your journal because our original research provides information that could be used as basis for future multicenter studies on this topic.
This manuscript has not been published or presented elsewhere in part or in entirety and is not under consideration by another journal. The study design was approved by the appropriate ethics review board. We have read and understood your journal’s policies, and we believe that neither the manuscript nor the study violates any of these. There are no conflicts of interest to declare.
Thank you for your consideration. I look forward to hearing from you.
Sincerely,
Tomoaki Ishigami, MD., PhD.,
Associate Professor
Department of Medical Science and Cardio-Renal Medicine
Yokohama City University Graduate School of Medicine
3-9, Fukuura, Kanazawa-Ku, YOKOHAMA, Kanagawa, Japan
TEL:+81-45-787-2635 ext 6312
FAX:+81-45-701-3738
Mail Address:tommmish@hotmail.com/tommmish@yokohama-cu.ac.jp
Dear Editors and Reviewers:
Thank you for your letter and for the reviewers' comments concerning our manuscript entitled "The types and proportions of commensal microbiota have predictive value in coronary heart disease" (ID: jcm-1158947). Those comments are all valuable and very helpful for revising and improving our paper, as well as the important guiding significance to our researches. We have studied comments carefully and have made correction which we hope meet with approval. Revised portion are marked in red in the paper. The main corrections in the paper and the responds to the reviewer's comments are as flowing:
Responds to the reviewer's comments:
Reviewer #1:
- Response to comment: Including “coronary artery disease” in your search for articles would have probably increased the number of results. In your methodology section you describe that the search was conducted to include “coronary heart disease”. This is confusing since in Table 1 most studies. The search strategy only includes one date, not a range. Does this mean that all studies in history up until March 2020 were included?
Response: According to the PICO criteria, in order to retrieve all relevant documents as much as possible up until March 2020, we tried a variety of terms combinations. When searching in the PubMed database, we finally chose the “Coronary Disease” and “Gastrointestinal Microbiome” as MeSH terms with the most search results. Of course, the entry terms also include “Coronary Heart Diseases”. When searching in the Embase database, “Coronary Disease” and “Gastrointestinal Microbiome” are both explosion terms of “Coronary Artery Disease” and “Intestine Flora”. Therefore, in order to avoid having no all-sided retrieval result, the descriptor terms we searched are “Coronary Artery Disease” and “Intestine Flora”. We are very sorry for our negligence of writing, and we have made correction according to the Reviewer's comments (Line 51-53, the statements of “We used broad search terms describing aspects of ‘commensal microbes’ and ‘CHD’." were corrected as “We used broad search terms describing aspects of ‘Gastrointestinal Microbiome’ and ‘Coronary Disease’.). Please refer to the attachment for more details about search strategy.
- Response to comment: The methodology of this study does not include any hypothesis, nor statistics. In page 2 lines 70-78, it is described that the 16 selected studies were sorted and classified according to the degree to which the populations were increased or decreased. I am not sure what this means, how it was determined, or what is the significance of this.
Response: As reviewer pointed out that we did not use statistic in this study, because the studies we searched all used RNA/DNA sequencing and without p value. These studies investigated the proportions and changes of commensal microbes, also including the increase or decrease in various types of commensal microbes. It is known that commensal microbes play a very important role in CHD whether increase or decrease, therefore, we sorted out the micorbiota shared among at least two papers regardless of increase or decrease in commensal microbes at first (table 2 and 3). And then, in order to accurately definitude the relationship between commensal microbes’ changes and CHD, we classified the changes of commensal microbes according to increases or decreases in blood, fecal and plaque. However, we found that some types of commensal microbes in fecal were increased in some studies whereas decreased in others. As we said in the discussion, this is maybe due to that several differences in the methodology of these studies, such as the region of genetic quantification, which could have influenced the results or may even have been significant sources of potential inaccuracy. In addition, the types and proportions of commensal microbiota could also have been influenced by several patient characteristics, such as diet (including regional differences). Therefore, in order to pick out the commensal microbes which could have predictive value more precisely, we further summarized the commensal microbes that only increased or decreased in fecal shared among at least two papers.
- Response to comment: Page 2, Line 82. States that 163 studies were excluded because they were “repetitive”. No explanation regarding what the authors mean by repetitive. It seems like a very large number of studies excluded.
Response: We are very sorry for no explanation on this part, line 82 is means that we deleted the duplicate literature (total 163 papers) through the EndNote software.
- Response to comment: Page 5, Line 136. It is not clear what the authors mean by comparisons of microbial communities between different sources. I am assuming the authors were looking to see if the same bacterial species were present in blood/feces, etc. However, this lacks scientific methodology. Therefore, I am not confident on the results and discussion that follows. Too many assumptions without supporting data.
Response: Our previous studies suggest that commensal microbes or commensal microbe-derived molecules can readily enter the bloodstream and exert systemic effects, which affecting B cell populations and then participate in atherosclerosis development. This comparison of blood, gut, and atherosclerotic plaque samples was necessary to identify members of the normal microbiome that may translocate from one body habitat to another where they may contribute to disease. As reviewer pointed out that we lacks the traditional statistical methods. However, we picked out the repeated commensal microbes through literature screening, and our results could be valuable information for future studies.
Special thanks to you for your good comments.
Reviewer 2 Report
Thank you for giving me the opportunity to review the manuscript entitled “The types and proportions of commensal microbiota have predictive value in coronary heart disease”. This study addresses an interesting link between the specific types of commensal microbiota and the occurrence of coronary heart disease (CHD). The manuscript is a systematic literature review including 16 papers (2,210 patients) with a prospective design.
The topic of this study is of interest but the review is superficial. First please formulate the aim of the study correctly. The first part is the aim indeed but the second part seems to be a summary of findings, but it is actually not supported by provided results.
The aim of the study was to confirm the types and proportions of commensal microbiota in people wt CHD? And I seem that you realised this aim in their paper. But how did you approach investigating the predictive value? Was there enough data in the literature on this topic? How was this aim realized?
Next, the synthesis of data is very superficial. Can I ask you to develop a table with a list of included studies and compare between what was found in the patients with CHD with the findings in patients without CHD? The comparative longitudinal analysis is missing to be able to predict any changes. Also, please present the characteristics of populations included in the study. We know that age can change gut bacteria so the reader should know the age of the analysed populations. What other factors were found in the included studies that affect gut microbiota?
Finally, I have also some concerns about methodology.
- The supplementary material is missing, so it is not possible to evaluate the search strategy. I am not able to comment on this part of the search; however, strings like ‘predictive value’ are not mentioned in the text.
- A flow diagram should be developed according to PRISMA.
- Also, the study selection criteria are blurred. Could you please elaborate on them in line with PICO criteria?
Author Response

(The authors gave the same response as above.)

Reviewer 3 Report
Authors performed a systematic review of prospective observational studies that correlate the commensal microbiota and CHD, in order to investigate the types and the cross-presence of specific commensal microbiota between atherosclerotic plaques, blood and gut. Even if I think the topic is interesting, I have some doubts about the analyses performed. I think is important to evaluate the comparability between the populations selected in the 16 publications included. Authors, in the study limitations, talked about diet, but I think is also more important to compare criteria like the gender, age, smoking, eventual treatments ongoing, stage of the pathology or similars that were used as selection criteria in the different papers. These parameters could strongly impact on the microbiota and, thus, influence on the presence or not in a specific compartment. I suggest to Authors to evaluate this aspect.
In addition, I suggest to add another table "Microbiota commonly identified in blood samples" as already done for plaque and fecal samples.
Author Response
Dear Editor:
I wish to submit an revised manuscript for publication in the Journal of Clinical Medicine, titled “The types and proportions of commensal microbiota have predictive value in coronary heart disease” The paper was coauthored by Lin Chen , Tomoaki Ishigami , Hiroshi Doi, Kentaro Arakawa and Kouichi Tamura.
We performed point by point responses to the reviewer’s comments as followings;
We believe that our study makes a significant contribution to the literature. Further, we believe that this paper will be of interest to the readership of your journal because our original research provides information that could be used as basis for future multicenter studies on this topic.
This manuscript has not been published or presented elsewhere in part or in entirety and is not under consideration by another journal. The study design was approved by the appropriate ethics review board. We have read and understood your journal’s policies, and we believe that neither the manuscript nor the study violates any of these. There are no conflicts of interest to declare.
Thank you for your consideration. I look forward to hearing from you.
Sincerely,
Tomoaki Ishigami, MD., PhD.,
Associate Professor
Department of Medical Science and Cardio-Renal Medicine
Yokohama City University Graduate School of Medicine
3-9, Fukuura, Kanazawa-Ku, YOKOHAMA, Kanagawa, Japan
TEL:+81-45-787-2635 ext 6312
FAX:+81-45-701-3738
Mail Address:tommmish@hotmail.com/tommmish@yokohama-cu.ac.jp
Dear Editors and Reviewers:
Thank you for your letter and for the reviewers' comments concerning our manuscript entitled "The types and proportions of commensal microbiota have predictive value in coronary heart disease" (ID: jcm-1158947). Those comments are all valuable and very helpful for revising and improving our paper, as well as the important guiding significance to our researches. We have studied comments carefully and have made correction which we hope meet with approval. Revised portion are marked in red in the paper. The main corrections in the paper and the responds to the reviewer's comments are as flowing:
Reviewer #3:
- Response to comment: In the study limitations, talked about diet, but I think is also more important to compare criteria like the gender, age, smoking, eventual treatments ongoing, stage of the pathology or similars that were used as selection criteria in the different papers. These parameters could strongly impact on the microbiota and, thus, influence on the presence or not in a specific compartment. I suggest to authors to evaluate this aspect.
Response: It is really true as reviewer suggested that parameters like the gender, age, smoking, eventual treatments ongoing, stage of the pathology could strongly impact on the microbiota, and compared these criteria are important. However, in the literature we searched, not all studies investigated these parameters, and the data about these is incomplete. Moreover, the bacteria we screened were those showed significant changes in the population with coronary artery lesions, and regardless of these parameters exist or not, we consider that these specific types of commensal microbiota could have more widespread predictive value for CHD. Our findings are based on the data from a limited number of studies this time, as reviewer's suggestion, we will investigate specific types of commensal microbiota, as well as factors that modulate or inhibit their activity in further studies.
- Response to comment: I suggest to add another table "Microbiota commonly identified in blood samples" as already done for plaque and fecal samples.
Response: As reviewer's suggestion, we have add the table of "Microbiota commonly identified in blood samples" according to the reviewer's suggestion, please refer to the attachment.
Special thanks to you for your good comments.
We tried our best to improve the manuscript and made some changes in the manuscript. These changes will not influence the content and framework of the paper. And here we did not list the changes but marked up using the “Track Changes” function in revised paper.
We appreciate for Editors/Reviewers' warm work earnestly, and hope that the correction will meet with approval.
Once again, thank you very much for your comments and suggestions.
Round 2
Reviewer 1 Report
Thank you for clarifying my prior questions. My concern with the present study is that conclusions are being made with very limited data and admittedly, no analysis (the science is not strong).
Author Response
Dear Editor:
I wish to submit a revised article for publication in the Journal of Clinical Medicine, "The types and proportions of commensal microbiota have predictive value in coronary heart disease" (ID: jcm-1158947). Those comments are all valuable and very helpful for revising and improving our paper, as well as the important guiding significance to our researches. We have studied comments carefully and have made correction which we hope meet with approval. Revised portion are marked in red in the paper. The main corrections in the paper and the responds to the reviewer's comments are as flowing:
Further, we believe that this paper will be of interest to the readership of your journal because our original research provides information that could be used as basis for future multicenter studies on this topic.
This manuscript has not been published or presented elsewhere in part or in entirety and is not under consideration by another journal. The study design was approved by the appropriate ethics review board. We have read and understood your journal’s policies, and we believe that neither the manuscript nor the study violates any of these. There are no conflicts of interest to declare.
Thank you for your consideration. I look forward to hearing from you.
Sincerely,
Tomoaki Ishigami, MD., PhD.,
Associate Professor
Department of Medical Science and Cardio-Renal Medicine
Yokohama City University Graduate School of Medicine
3-9, Fukuura, Kanazawa-Ku, YOKOHAMA, Kanagawa, Japan
TEL:+81-45-787-2635 ext 6312
FAX:+81-45-701-3738
Mail Address:tommmish@hotmail.com/tommmish@yokohama-cu.ac.jp
Reviewer #1
General Comments: Comments and Suggestions for Authors
Thank you for clarifying my prior questions. My concern with the present study is that conclusions are being made with very limited data and admittedly, no analysis (the science is not strong).
Response: We deeply appreciate your time. We would like to express our gratitude for reviewing our manuscript. According to the reviewer’s comment, we agreed that our manuscript have some limitation to conclude microbiological background for human atherosclerosis. Therefore, we revised our manuscript currently as followings to declare substantial limitations underlying our research.
“Study strengths and limitations
Although some studies investigating the relationship between commensal microbiota and CHD have been conducted previously [6], our synthetic analysis is the first to investigate the types of commensal microbiota that are commonly associated with CHD. Moreover, our findings revealed several specific types of commensal microbiota that commonly exist or coexist in the atherosclerotic plaque, blood, or feces of patients with CHD. This could be valuable knowledge for future studies investigation this association, as it reveals that the microbiota of the atherosclerotic plaque may, at least in part, be derived from the gut, and these specific types of commensal microbiota may have predictive value for CHD.
A limitation of this study was that only 16 relevant publications met the criteria for inclusion, with only three contributing to the comparisons of bacterial population in the atherosclerotic plaque, and only two studies contributing to that of the blood. This limited our ability to generate a precise descriptive summary of the ‘real-world’ changes in relative commensal microbiota populations. Furthermore, there were several differences in the methodology of these studies, such as the region of genetic quantification, which could have influenced the results or may even have been significant sources of potential inaccuracy. In addition, the types and proportions of commensal microbiota could also have been influenced by several patient characteristics, such as diet (including regional differences) [44]. The addition of new evidence to the field will significantly reduce the effects of these limitations in future analyses.
Further studies should investigate specific types of commensal microbiota, as well as factors that modulate or inhibit their activity. In addition, more information is required to verify the predictive value and mechanisms of commensal microbiota in CHD patients.” (page 9, paragraph 2)
Reviewer 2 Report
I do not see any answer to my previous comments. No changes were made to address my comments from the first round. If that is by mistake, please upload the correct responses and correct manuscript.
Author Response
Dear Editor:
I wish to submit a revised article for publication in the Journal of Clinical Medicine, "The types and proportions of commensal microbiota have predictive value in coronary heart disease" (ID: jcm-1158947). Those comments are all valuable and very helpful for revising and improving our paper, as well as the important guiding significance to our researches. We have studied comments carefully and have made correction which we hope meet with approval. Revised portion are marked in red in the paper. The main corrections in the paper and the responds to the reviewer's comments are as flowing:
Further, we believe that this paper will be of interest to the readership of your journal because our original research provides information that could be used as basis for future multicenter studies on this topic.
This manuscript has not been published or presented elsewhere in part or in entirety and is not under consideration by another journal. The study design was approved by the appropriate ethics review board. We have read and understood your journal’s policies, and we believe that neither the manuscript nor the study violates any of these. There are no conflicts of interest to declare.
Thank you for your consideration. I look forward to hearing from you.
Sincerely,
Tomoaki Ishigami, MD., PhD.,
Associate Professor
Department of Medical Science and Cardio-Renal Medicine
Yokohama City University Graduate School of Medicine
3-9, Fukuura, Kanazawa-Ku, YOKOHAMA, Kanagawa, Japan
TEL:+81-45-787-2635 ext 6312 FAX:+81-45-701-3738
Mail Address:tommmish@hotmail.com/tommmish@yokohama-cu.ac.jp
Reviewer #2
General Comments: Comments and Suggestions for Authors
I do not see any answer to my previous comments. No changes were made to address my comments from the first round. If that is by mistake, please upload the correct responses and correct manuscript.
Response: We deeply appreciate your time. We would like to express our gratitude for reviewing our manuscript. According to the reviewer’s comment, we need to express our sincere sorry about previous response. Currently, we replied appropriately to the reviewer’s comment as followings;
Reviewer #2:
Response to comment: How did you approach investigating the predictive value? Was there enough data in the literature on this topic? How was this aim realized?
Response: We are very sorry for our inaccurate description. As we said in the limitation, only 16 relevant publications met the criteria for inclusion at last, this limited our ability to generate a precise descriptive summary of the ‘real-world’ changes in relative commensal microbiota populations. However, we found key types of bacteria associated with CHD though literature screening, and we consider that these specific types of commensal microbiota have predictive value for CHD. Of course, our findings are based on the data from a limited number of studies, more prospective studies are needed to further investigate and verify predictive value of specific types of commensal microbiota in CHD.
Response to comment: The synthesis of data is very superficial. Can I ask you to develop a table with a list of included studies and compare between what was found in the patients with CHD with the findings in patients without CHD? The comparative longitudinal analysis is missing to be able to predict any changes. Also, please present the characteristics of populations included in the study. We know that age can change gut bacteria so the reader should know the age of the analysed populations. What other factors were found in the included studies that affect gut microbiota?
Response: Considering the reviewer's suggestion, we have add the list of included studies, please refer to the attachment for details about search literature. As reviewer said that age can change gut bacteria, however, the bacteria we screened were those showed significant changes in the population with coronary artery lesions, and regardless of the control group and age, we consider that these specific types of commensal microbiota could have more widespread predictive value for CHD. And as attachment shown, not all studies investigated the factors that affect the gut microbiome, therefore, we did not consider these factors this time.
Response to comment: about methodology. The supplementary material is missing, so it is not possible to evaluate the search strategy. I am not able to comment on this part of the search; however, strings like ‘predictive value’ are not mentioned in the text. A flow diagram should be developed according to PRISMA. Also, the study selection criteria are blurred. Could you please elaborate on them in line with PICO criteria?
Response: We are very sorry for our negligence of supplementary material. Please refer to the attachment for details about search strategy.
Special thanks to you for your good comments.
Reviewer 3 Report
I really thank Authors for the kind reply. However, I don't think modifications are sufficient.
For what concern the table on blood samples, I think will be more useful to add it in the manuscript for a complete and better comprehension of your analysis.
For what concern the characteristics of the population in the studies taken in consideration, I think that is fundamental to investigate them if Authors want to propose a particular type of commensal microbiota as a predictive value for CHD, in order to find a strong correlation not affected by any bias. Indeed, in absence of this analysis, the work can be cosidered as an indication of changes in the microbiota in the population with coronary artery lesions, but is not enough to consider this changes as reliable predictive values.
Author Response
Dear Editor:
I wish to submit a revised article for publication in the Journal of Clinical Medicine, "The types and proportions of commensal microbiota have predictive value in coronary heart disease" (ID: jcm-1158947). Those comments are all valuable and very helpful for revising and improving our paper, as well as the important guiding significance to our researches. We have studied comments carefully and have made correction which we hope meet with approval. Revised portion are marked in red in the paper. The main corrections in the paper and the responds to the reviewer's comments are as flowing:
Further, we believe that this paper will be of interest to the readership of your journal because our original research provides information that could be used as basis for future multicenter studies on this topic.
This manuscript has not been published or presented elsewhere in part or in entirety and is not under consideration by another journal. The study design was approved by the appropriate ethics review board. We have read and understood your journal’s policies, and we believe that neither the manuscript nor the study violates any of these. There are no conflicts of interest to declare.
Thank you for your consideration. I look forward to hearing from you.
Sincerely,
Tomoaki Ishigami, MD., PhD.,
Associate Professor
Department of Medical Science and Cardio-Renal Medicine
Yokohama City University Graduate School of Medicine
3-9, Fukuura, Kanazawa-Ku, YOKOHAMA, Kanagawa, Japan
TEL:+81-45-787-2635 ext 6312 FAX:+81-45-701-3738
Mail Address:tommmish@hotmail.com/tommmish@yokohama-cu.ac.jp
Reviewer #3
General Comments: Comments and Suggestions for Authors
I really thank Authors for the kind reply. However, I don't think modifications are sufficient. For what concern the table on blood samples, I think will be more useful to add it in the manuscript for a complete and better comprehension of your analysis.
Response: We deeply appreciate your time. We would like to express our gratitude for reviewing our manuscript. According to the reviewer’s comment, we revised Table 6 as an attachment of current our manuscripts as followings;
Comments :For what concern the characteristics of the population in the studies taken in consideration, I think that is fundamental to investigate them if Authors want to propose a particular type of commensal microbiota as a predictive value for CHD, in order to find a strong correlation not affected by any bias. Indeed, in absence of this analysis, the work can be cosidered as an indication of changes in the microbiota in the population with coronary artery lesions, but is not enough to consider this changes as reliable predictive values.
Response: We deeply appreciate your time. We would like to express our gratitude for reviewing our manuscript. According to the reviewer’s comment, we agreed that our manuscript have some limitation to conclude microbiological background for human atherosclerosis. Therefore, we revised our manuscript currently as followings to declare substantial limitations underlying our research.
“Study strengths and limitations
Although some studies investigating the relationship between commensal microbiota and CHD have been conducted previously [6], our synthetic analysis is the first to investigate the types of commensal microbiota that are commonly associated with CHD. Moreover, our findings revealed several specific types of commensal microbiota that commonly exist or coexist in the atherosclerotic plaque, blood, or feces of patients with CHD. This could be valuable knowledge for future studies investigation this association, as it reveals that the microbiota of the atherosclerotic plaque may, at least in part, be derived from the gut, and these specific types of commensal microbiota may have predictive value for CHD.
A limitation of this study was that only 16 relevant publications met the criteria for inclusion, with only three contributing to the comparisons of bacterial population in the atherosclerotic plaque, and only two studies contributing to that of the blood. This limited our ability to generate a precise descriptive summary of the ‘real-world’ changes in relative commensal microbiota populations. Furthermore, there were several differences in the methodology of these studies, such as the region of genetic quantification, which could have influenced the results or may even have been significant sources of potential inaccuracy. In addition, the types and proportions of commensal microbiota could also have been influenced by several patient characteristics, such as diet (including regional differences) [44]. The addition of new evidence to the field will significantly reduce the effects of these limitations in future analyses.
Further studies should investigate specific types of commensal microbiota, as well as factors that modulate or inhibit their activity. In addition, more information is required to verify the predictive value and mechanisms of commensal microbiota in CHD patients.” (page 9, paragraph 2)